Distribution of rhizosphere fungi of Kobresia humilis on the Qinghai-Tibet Plateau

Guo Jing 1
Xie Zhanling 2 3 xiezhanling2012@126.com
Meng Qing 2 3
Xu Hongyan 4
Peng Qingqing 2 3
Wang Bao 2
Dong Deyu 2
Yang Jiabao 2
Jia Shunbin 5
1 School of Ecology and Environmental Science, Qinghai University of Science and Technology , Xining , China
2 College of Ecological and Environment Engineering, Qinghai University , Xining , China
3 State Key Laboratory Breeding Base for Innovation and Utilization of Plateau Crop Germplasm, Qinghai University , Xining , China
4 Academy of Agriculture and Forestry Sciences, Qinghai University , Xining , China
5 Department of Ecological Restoration at Qinghai Grassland Station , Xining , China
Passari Ajit
Electronic publication date: 2024 Feb 20
Publication date: 2024
Volume: 12
Electronic Location ID: e16620
Received 2023 May 23; Accepted 2023 Nov 16
Copyright: © 2024 Guo et al.
Copyright year: 2024
Copyright holder: Guo et al.
License: This is an open access article distributed under the terms of the Creative Commons Attribution License, which permits unrestricted use, distribution, reproduction and adaptation in any medium and for any purpose provided that it is properly attributed. For attribution, the original author(s), title, publication source (PeerJ) and either DOI or URL of the article must be cited.
License URL: https://creativecommons.org/licenses/by/4.0/

Keywords: Kobresia humilis, Fungal distribution, Rhizosphere, Qinghai-Tibet Plateau, Long-term evolutionary process

Funding: First Batch of Central Forestry and Grassland Ecological Protection and Restoration Funds in 2021 2021-87 Natural Science Planning Project of Qinghai Province 2024-0204-SFC-0013 This research was funded by the first batch of central forestry and grassland ecological protection and restoration funds in 2021 (2021-87) and supported by funding from the Natural Science Planning Project of Qinghai Province (2024-0204-SFC-0013). The funders had no role in study design, data collection and analysis, decision to publish, or preparation of the manuscript.

==============================
Kobresia humilis is a major species in the alpine meadow communities of the Qinghai-Tibet Plateau (QTP); it plays a crucial role in maintaining the ecological balance of these meadows. Nevertheless, little is known about the rhizosphere fungi associated with K. humilis on the Qinghai Tibet Plateau. In this study, we used Illumina Miseq to investigate the fungal diversity, community structure, and ecological types in the root and rhizosphere soil of K. humilis across eight areas on the QTP and analyzed the correlation between rhizosphere fungi of K. humilis and environmental factors. A total of 19,423 and 25,101 operational taxonomic units (OTUs) were obtained from the roots and rhizosphere soil of K. humilis. These were classified into seven phyla, 25 classes, 68 orders, 138 families, and 316 genera in the roots, and nine phyla, 31 classes, 76 orders, 152 families, and 407 genera in the rhizosphere soil. There were 435 and 415 core OTUs identified in root and rhizosphere soil, respectively, which were categorized into 68 and 59 genera, respectively, with 25 shared genera. Among them, the genera with a relative abundance >1% included Mortierella, Microscypha, Floccularia, Cistella, Gibberella, and Pilidium. Compared with the rhizosphere soil, the roots showed five differing fungal community characteristics, as well as differences in ecological type, and in the main influencing environmental factors. First, the diversity, abundance, and total number of OTUs in the rhizosphere soil of K. humilis were higher than for the endophytic fungi in the roots by 11.85%, 9.85%, and 22.62%, respectively. The composition and diversity of fungal communities also differed between the eight areas. Second, although saprotroph-symbiotrophs were the main ecological types in both roots and rhizosphere soil; there were 62.62% fewer pathotrophs in roots compared to the rhizosphere soil. Thirdly, at the higher altitude sites (3,900–4,410 m), the proportion of pathotroph fungi in K. humilis was found to be lower than at the lower altitude sites (3,200–3,690 m). Fourthly, metacommunity-scale network analysis showed that during the long-term evolutionary process, ZK (EICZK = 1) and HY (EICHY = 1) were critical sites for development of the fungal community structure in the roots and rhizosphere soil of K. humilis, respectively. Fifthly, canonical correspondence analysis (CCA) showed that key driving factors in relation to the fungal community were longitude (R2 = 0.5410) for the root community and pH (R2 = 0.5226) for the rhizosphere soil community. In summary, these results show that K. humilis fungal communities are significantly different in the root and rhizosphere soil and at the eight areas investigated, indicating that roots select for specific microorganisms in the soil. This is the first time that the fungal distribution of K. humilis on the QTP in relation to long-term evolutionary processes has been investigated. These findings are critical for determining the effects of environmental variables on K. humilis fungal communities and could be valuable when developing guidance for ecological restoration and sustainable utilization of the biological resources of the QTP.

Introduction

The Qinghai-Tibet Plateau (QTP) is regarded as the Earth’s third pole, with a distinct ecological type (Xu et al., 2011). Alpine meadows, which cover one-third of the QTP, experience a low annual temperature, a large diurnal temperature variation, and only a minor seasonal temperature change (Wang et al., 2008; Schleuss et al., 2015; Ma et al., 2021). Degradation of alpine Kobresia meadow is widespread and has recently expanded over the QTP because of overgrazing and climate change (Li et al., 2015). Kobresia is an integral part of the alpine meadow ecosystem, which is widely distributed in the east and southwest of the QTP, including the area around Qinghai Lake and the Himalayan-Hengduan mountain ranges (Hu et al., 2019). There are more than 79 species of Kobresia in China, 17 of which are found in the alpine Kobresia meadows of the QTP: K. macrantha, K. stolonifera, K. fragilis, K. kansuensis, K. royleana, K. minshanica, K. pygmaea, K. graminifolia, K. robusta, K. pusilla, K. filifolia, K. cuneata, K. schoenoides, K. tibetica, K. humilis, K. capillifolia, and K. littledalei (Rajbhandari & Ohba, 1988). Among them, K. humilis is an endemic and widely distributed plant of the QTP’s alpine Kobresia meadows (Wu et al., 2011).

Kobresia humilis, a perennial with a well-developed root system, belongs to the sedge family (Zhu et al., 2004). It is a constructive species in alpine Kobresia meadow, distributed in alpine areas of six provinces in China Tibet, Gansu, Xinjiang, Yunnan, Sichuan, and Qinghai at an altitude of 2,100–4,500 m (Zheng et al., 2009; Zhang, 2003). It grows well in harsh alpine environmental conditions due to its effective resistance to cold, drought, radiation, wind, and exposure (Li et al., 2006). In the process of long-term adaptation to cold and resistance to grazing by livestock, K. humilis has developed the ability to survive challenges to morphology and nutrition and is resistant to drought and trampling (Wang et al., 2022a; Cao et al., 2010). Furthermore, K. humilis contains high levels of crude proteins and fats and is favored by livestock (Han, Ben & Shi, 1988). Most of the current research on K. humilis focuses on its root biomass, production, the uptake diversity of soil nitrogen nutrients, and the influence of climatic changes on distribution patterns (Wu et al., 2011; Wang et al., 2012; Hu et al., 2019). To date, no research has been conducted on endophytic fungi and fungal communities in the rhizosphere soil of Kobresia humilis.

Plant roots and their surrounding rhizosphere soil are crucial habitats for fungi. Roots secrete substances into the soil to alter its physical and chemical environment, which then influences the species, quantity, and composition of rhizosphere fungi, resulting in a change in soil community (Hong et al., 2018; Chen et al., 2020; Shi et al., 2021; Aleklett et al., 2022). Fungi degrade complex compounds more efficiently; fungi provide nutrition to plants (Guan et al., 2018; Sun et al., 2021) but are more sensitive to changes in the soil environment, so their dynamics can indicate soil ecological changes (Shi et al., 2021). The rhizosphere soil is a complex environment harboring diverse organisms potentially beneficial to plants (Inselsbacher & Näsholm, 2012; Arslan et al., 2020). Fungal communities in the rhizosphere soil are essential in biogeochemical cycles, plant growth, organic matter decomposition, disease suppression, and pathogen antagonism (Raaijmakers et al., 2009). Fungal communities vary from plant to plant because of the root exudates preferred by different soil microorganisms (Klaubauf et al., 2010). Physiochemical properties like soil texture and enzyme activities directly affect the rhizosphere fungal network (Arslan et al., 2020).

This study examined roots and rhizosphere soil of K. humilis from eight areas on the QTP. We explored the diversity, community structure, and ecological function of endophytic fungi and rhizosphere soil fungi of K. humilis and also investigated the correlation between endophytic fungi and rhizosphere soil fungi with environmental factors to explore the potential influence of fungal community on the establishment of K. humilis as a dominant species in alpine meadow ecosystems. In addition, the current study provides valuable information to apply when protecting and sustainably utilizing these alpine meadow resources.

Materials and Methods

Sample collection

The K. humilis root and rhizosphere soil samples were collected from the following regions from July to August 2020: Haiyan (HY, 100°48′13″E, 36°59′44″N), ZeKu (ZK, 101°28′1″E, 35°3′26″N), XingHai (XH, 99°47′5″E, 35°50′16″N), TianJun (TJ, 99°8′17″E, 37°13′40″N), QingShui river (QSH, 97°9′24″E, 33°49′59″N), ZhenQin (ZQ, 97°18′59″E, 33°23′34″N), MaQin (MQ, 100°21′5″E, 34°25′35″N), GanDe (GD, 99°41′5″E, 33°47′56″N), and Qinghai-Tibet Plateau, China (Fig. 1; Table S1). All root and rhizosphere soil samples were collected from wild K. humilis at the eight sites, with altitudes ranging from 3,200 to 4,410 m a.s.l. Each sample point was set with ten repeats, and the distance between the each repeats was above 20 m. First, rhizosphere soil was collected according to Chen et al. (2020). Then, the rhizosphere soil samples were divided into two parts: one for determining soil physicochemical properties and one for analyzing fungal communities. All root and rhizosphere soil samples were placed in sterile autoclaved bags, labeled, and transported to the laboratory in an icebox and stored at −80 °C whilst awaiting further processing.

Figure 1 Geographic distribution of sampling sites across the eight regions on Qinghai-Tibet Plateau.

Determination of rhizosphere soil physicochemical data

After air-drying the rhizosphere soil samples and passing them through a 2-mm sieve, the total nitrogen (TN), total phosphorous (TP), and total organic carbon (TOC) were measured using standard soil testing procedures. Analyses were conducted by the Institute of Soil and Fertilizer, Qinghai Academy of Agriculture and Forestry Sciences. Soil moisture content was determined using the gravimetric method. Soil pH was determined using a pH meter and a soil suspension with a soil-water ratio of 1.0: 2.5 (He et al., 2021). The daily meteorological data (Max T: daily maximum temperature; A T: average temperature; Min T: daily minimum temperature; SD: sunshine duration; DTR: daily temperature range; ARH: average relative humidity; WS: wind speed.) for July of 2020 was obtained from Qinghai Meteorological Bureau.

DNA extraction, PCR, and Illumina MiSeq sequencing

Total genomic DNA from the K. humilis root and rhizosphere soil samples was extracted using the CTAB method (Chen et al., 2018) and an Ezup Column Soil DNA Purification Kit (Sangon Biotech, Shanghai, China), respectively. DNA purity was quantified by a NanoDrop spectrophotometer and checked by 1% agarose gel electrophoresis. PCR was then performed using specific barcode primers, and the reactions were conducted using Taq PCR Master Mix (Sangon Biotech, Shanghai, China). The fungal ITS region was analyzed using fungal-specific primers ITS1 (CTTGGTCATTTAGAGGAAGTAA) and ITS4 (TCCTCCGCTTATTGATATGC). Each 25 μL volume of the PCR reaction solution contained 12.5 μL of 2× Taq PCR Master Mix, 1.0 μL of each primer, 1.0 μL of template DNA, and 9.5 μL of ddH2O. Amplifications were performed using the following protocol: initial denaturation of 1 min at 94 °C; 35 cycles of 15 s at 94 °C, 15 s at 58 °C, and 1 min at 72 °C. Final elongation was performed at 72 °C for 5 min. The success of the PCR products was tested in 1% agarose gel and then purified using a SanPrep Column DNA Gel Extraction Kit (Sangon Biotech, Shanghai, China). The qualified samples were sent to BioBit Biotech Inc. for sequencing. The DNA library used a TruSeq® DNA PCR-free Sample Preparation Kit and Qubit and qPCR for quantification. After the library was qualified, the Illumina sequencing platform was used for sequencing.

Bioinformatics analyses

In order to improve the reliability of the data processing, sequence reads were assigned to each sample based on their unique barcode and truncated by removing the barcode and primer sequences. Paired-end reads from the original DNA fragments were merged using FLASH (v1.2.7, http://ccb.jhu.edu/software/FLASH/), and QIIME (v1.9.0, http://qiime.org/scripts/split_libraries_fastq.html) was used to demultiplex FASTQ files, filter chimeric sequences, and treat sequence ends to remove low-quality regions. Singletons were discarded and the dataset was dereplicated, then the remaining sequences clustered into operational taxonomic units (OTUs) at a 97% sequence identity using USEARCH version 8.0 (Wang et al., 2020). The representative sequences for each OTU were screened and taxonomically aligned using UNITE for fungi. All sequences in the current study are stored in the sequence reading Archive (SRA) of the NCBI database, with biological project ID PRJNA965955 and accession numbers SAMN34510363, and SAMN34510364.

Statistical analysis

The OTU abundance data were normalized using a standard sequence number corresponding to the sample with the fewest sequences. Subsequent analyses of diversity were performed based on the normalized data. First, the community composition of each treatment was analyzed in relation to the five classification levels of phylum, class, order, family, and genus. Next, the community composition dissimilarities (Bary-Curtis distance) of fungi were visualized with non-metric multidimensional scaling (NMDS) using Omicshare tools (https://www.omicshare.com/tools). Next, three diversity indices were calculated to examine the species richness and diversity of the samples: Chao 1, Shannon, and Simpson. These indices were calculated for all samples with QIIME, and the results were plotted with R software (Lai et al., 2022). Rarefaction curves were generated based on these metrics. To compare species-environment correlations, canonical correspondence analysis (CCA) was performed (Sun et al., 2021). Finally, the functional group of each OTU was assigned using FunGuild.

Results

Analysis of fungal diversity in the rhizosphere of K. humilis

The total DNA was sequenced to obtain 7,723–43,343 ITS sequences for roots and 14,190–36,623 for the rhizosphere soil (Fig. S1). The total number of effective fungal sequences from the isolated DNA was 200,980 for roots and 179,606 for the rhizosphere soil, and the average number of sequences was 25,122 and 22,450, respectively (Table S2). The average sequence lengths were 238 for roots and 240 for the rhizosphere soil. The optimized sequences were screened after cluster analysis. They were divided into 19,423 operational taxonomic units (OTUs) for roots and 25,101 for the rhizosphere soil at the 97% sequence similarity level. The total number of OTUs in the rhizosphere soil was 22.62% higher than in the roots (Fig. S2). The rarefaction curve showed a levelling off with increasing number of sequences, indicating that the data were valid and the sequencing depth was sufficient (Fig. S3).

Alpha diversity represents within-community fungal diversity, allowing a comparison of the diversities for K. humilis roots and the rhizosphere soil between sites (Fig. 2). According to analysis using the Shannon index, the fungal species richness was highest (7.12) for the roots at GD, followed by QSH (6.65), ZQ (6.72), ZK (6.31), HY (6.07), XH (5.36), MQ (4.85), and TJ (4.53) (Fig. 2A). The Chao1 index values were GD (1,469.96) > QSH (1,401.82) > ZQ (1,274.85) > ZK (1,265.41) > HY (886.75) > XH (859.21) > TJ (599.09) > MQ (536.71) (Fig. 2B). Thus, root fungal community diversity was greatest at GD. As shown in Fig. 2C, MQ had the lowest Simpson index, indicating a higher level of fungal community diversity. In contrast, HY had the greatest Simpson index, indicating less fungal community diversity than other areas. In terms of the average value of Chao’s index for the fungal community in the rhizosphere soil, the ranking was QSH (2,195.68) > HY (2,132.10) > MQ (926.71) > XH (907.59) > ZK (870.24) > TJ (766.21) > ZQ (755.41) > GD (646.02). Thus, fungal community diversity in the rhizosphere soil was greatest at QSH. The Chao1 diversity and Shannon index values of the QSH rhizosphere soil fungi were 2,195.68 and 8.86, followed by those at HY, which were 2,132.10 and 8.43, respectively, while the GD rhizosphere soil had the lowest values of 646.02 and 4.27 (Figs. 2D–2E).

Figure 2 Fungal community diversity index of root and rhizosphere soil.

In order to better display the distance relationship between multiple samples, the fungal β diversity was further assessed based on an unweighted UniFrac distance matrix (Costa et al., 2021). The root samples from TJ, QSH, ZQ, MQ, and GD regions were similar in community structure (Fig. 3A). However, the similarity between the rhizosphere soil samples was low. The community structure was different in the eight areas (Fig. 3B). The fungal community composition of both roots and rhizosphere soil varied between different samples, as illustrated by the nonmetric multidimensional scaling (NMDS) (Figs. 3C and 3D). Moreover, the community composition was affected by the sampling sites.

Figure 3 Unweighted UniFrac clustering and NMDS of microbial communities in K. humilis root and rhizosphere soil.

Distribution of fungi in the rhizosphere of K. humilis

In the root samples, the operational taxonomic units (OTUs) detected from the eight areas belonged to seven phyla, 25 classes, 68 orders, 138 families, and 316 genera. The root samples contained seven phyla and other phyla that accounted for more than 99% of the fungal sequences (Fig. 4A). The seven determined phyla were Ascomycota, Basidiomycota, Mortierellomycota, Olpidiomycota, Rozellomycota, Chytridiomycota, and Glomeromycota. The abundance of Ascomycota was highest in the HY (64.34%), ZK (62.04%), and XH (64.28%) root samples; and the abundance of Mortierellomycota was highest in the TJ (66.00%), QSH (54.15%), ZQ (61.53%), MQ (67.07%), and GD (50.06%) root samples. To better illustrate the distribution, the community Column chart shows only the ten most abundant fungi at the order level. Fungi in the roots from the eight locations were mainly members of the following ten orders: Agaricales, Hypocreales, Helotiales, Mortierellales, Auriculariales, Pleosporales, Xylariales, Minutisphaerales, Chaetothyriales and Sebacinales. Agaricales were most abundant in HY (21.41%), ZK (24.92%) and XH (5.58%). However, Mortierellales were most abundant in TJ (66.00%), QSH (54.15%), ZQ (61.53%), MQ (67.08%), and GD (50.07%) (Fig. 4B). Fungal community composition was further analyzed at the genus level. The following genera had a relative abundance >1%: Mortierella, Microscypha, Cistella, Floccularia, Nectria, Gibberella, Calyptella, Psathyrella, Auricularia, and Scytalidium. The most abundant fungi at the genus level in TJ, QSH, ZQ, MQ, and GD (Fig. 4C) were Mortierella, ranging in relative abundance from 4.13% to 67.07%.

Figure 4 Fungal community patterns in samples of root and rhizosphete soil.

For the fungal communities in the rhizosphere soil samples, OTUs from eight areas were found to belong to nine phyla, 31 classes, 76 orders, 152 families, and 407 genera. Figure 4D depicts the phylum-level composition and abundance of the fungal community. The nine phyla identified were: Ascomycota, Basidiomycota, Mortierellomycota, Chytridiomycota, Glomeromycota, Mucoromycota, Olpidiomycota, Rozellomycota, and Zoopagomycota. Ascomycota and Basidiomycota accounted for the largest proportion of roots and rhizosphere soil samples in HY, ZK, and XH. However, Ascomycota and Mortierellomycota accounted for the largest proportion of roots and rhizosphere soil samples in TJ, QSH, ZQ, MQ, and GD. At the order level, the fungi in the rhizosphere soil of the eight areas were mainly members of the following: Pleosporales, Agaricales, Mortierellales, Capnodiales, Helotiales, Hypocreales, Onygenales, Pezizale, Xylariales and Thelebolales (Fig. 4E). There were significant differences in the community distribution of fungi in the rhizosphere soils of different regions. Pleosporales were the most abundant fungi in the rhizosphere soils of HY, ZK, and XH, accounting for 18.01%, 31.45%, and 34.42%, respectively. However, Mortierellales was a dominant fungal order in the rhizosphere soils of TJ, QSH, ZQ, MQ, and GD; indeed abundance was the highest in GD (89.86%). In order to see the differences between the samples from the eight areas, the most abundant genera found in all samples were compared (Fig. 4F). Genera with a relative abundance >1% were Mortierella, Gibberella, Preussia, Didymella, Septoria, Floccularia, Zymoseptoria, Microdochium, Mycosphaerella, and Gymnoascus.

The OTUs identified in all analyzed samples were considered core OTUs (Lei et al., 2020). In total, 435 core OTUs were identified in the roots, and 415 in the rhizosphere soil at all sites (Fig. 4G-H). There were 371, 991, 418, 354,737, 481, 209, and 310 fungal OTUs unique to HY, ZK, XH, TJ, QSH, ZQ, MQ, and GD roots, respectively (Fig. 4G), and 1,290, 299, 685, 602, 862, 502, 277, and 212, respectively, in the rhizosphere soil (Fig. 4H). In root and rhizosphere soil, there were 68 and 59 identified genera, respectively, with 25 shared genera (Fig. 4I). The genera with a relative abundance >1% were Mortierella, Microscypha, Floccularia, Cistella, Gibberella, and Pilidium.

Typification of the fungal community in the rhizosphere of K. humilis

When evaluating the ecological functions of the fungal community found in the roots and rhizosphere soil of K. humilis, a different abundance profile was found for different sites. Based on the trophic mode of the fungal OTUs, nine functional roles were identified: symbiotroph, saprotroph-symbiotroph, saprotroph, pathotroph-symbiotroph, pathotroph-saprotroph-symbiotroph, pathotroph-saprotroph, pathotroph, pathogen- saprotroph-symbiotroph, and unknown (Figs. 5A and 5B). Saprotroph-symbiotroph was the main ecological function of fungi in the roots (relative abundance = 50.09%) and rhizosphere soil (relative abundance = 25.27%). There were few pathotrophs (relative abundance = 3.93%%) in roots, but more pathotrophs (relative abundance = 12.12%) and pathotroph-saprotroph (relative abundance = 11.43%) in the rhizosphere soil. Moreover, saprotrophs dominated the root and rhizosphere soil in HY, ZK, and XH. Meanwhile, the saprotroph-symbiotroph mode was abundant in the root and rhizosphere soil in TJ, QSH, ZQ, MQ, and GD. However, the unknown category meant that the functional roles of many fungi present were not classified. Studies have shown that FUNGuild can provide important information about fungal functions, but there is still work to be done to improve the overall information (Yao et al., 2020).

Figure 5 The ecological function prediction of fungal community in root and rhizosphere soil of K. humilis.

The detailed prediction and classification of species’ ecological functions in root and rhizosphere soil samples are shown in the cluster heatmap (Figs. 5C and 5D). Endophyte-Litter Saprotroph-Soil Saprotroph-Undefined Saprotroph was identified as the primary ecological function. Mortierella was the dominant taxon with a relative abundance of 39.27% and 29.27% in root and rhizosphere soils, respectively. The relative abundance of Nectria with the ecological functional modes of Animal Pathogen-Endophyte-Fungal Parasite-Lichen Parasite-Plant Pathogen-Wood Saprotroph was 2.41%. The relative abundance of Preussia with the ecological function of Dung Saprotroph-plant Saprotroph in rhizosphere soil was 4.18%. Other ecological functions, such as Ectomycorrhiza, which promotes plant growth and stress resistance, and Wood Saprotroph and Dung Saprotroph, which break down organic matter and promote nutrient cycling, were comparatively underrepresented in the root and rhizosphere soil samples.

Correlation between endophytic fungi/rhizosphere soil fungi of K. humilis and environmental factors

Figure 6A shows the metacommunity-scale network for root fungal taxa. There were 132 nodes and 646 edges; the average network distance between all pairs of nodes (average path length) was 2.132 edges, and the modularity index was 0.276. The entire network can be divided into two modules. ZK was the most important site based on the eigenvector centrality (EICZK = 1), followed by XH, QSH, ZQ, and HY (EICXH = 0.9967; EICQSH = 0.9664; EICZQ = 0.9625; EICHY = 0.9562). TJ, GD, and MQ had low centrality (EICTJ = 0.8889; EICGD = 0.8831; EICMQ = 0.8437) (Fig. 6A). Within the network topology, 56 fungi were located in central topological positions (Table S3). Figure 6B shows the metacommunity-scale network for rhizosphere soil fungal taxa. There are 148 nodes and 713 edges; the average network distance between all pairs of nodes (average path length) was 2.145 edges, and the modularity index was 0.198. We found that HY was the critical site based on the eigenvector centrality (EICHY = 1), followed by QSH, TJ, MQ, and XH (EICQSH = 0.9810; EICTJ = 0.9448; EICMQ = 0.9234; EICXH = 0.9123). ZQ, ZK, and GD had low centrality (EICZQ = 0.8311; EICZK = 0.7741; EICGD = 0.6542) (Fig. 6B). Within the network topology, 47 fungi were found in central topological positions (Table S4).

Figure 6 Composition of fungal taxa (family level) of root (A) and rhizosphere soil (B) in the metacommunity-scale network.

For root samples, the first CCA axis (CCA1) explained 48.52% of the variation (Fig. 7A). The correlation coefficients with altitude, latitude, longitude, total nitrogen (TN), total phosphorus (TP), soil organic carbon (SOC), soil moisture content (SMC), pH, daily maximum temperature (Max T), average temperature (AT), daily minimum temperature (Min T), sunshine duration (SD), daily temperature range (DTR), precipitation, average relative humidity (ARH) and wind speed (WS) (Table S5) were 0.4422, −0.0628, −0.981, 0.3806, 0.4443, 0.5249, 0.7319, −0.0508, −0.562, −0.659, −0.7365, −0.4633, 0.8106, −0.7218, −0.9969, and 0, respectively. The second axis (CCA2) explained 37.79% of the variation, and the correlation coefficients with altitude, latitude, longitude, TN, TP, SOC, SMC, pH, Max T, AT, Min T, SD, DTR, precipitation, ARH, and WS were −0.8969, 0.998, 0.1939, 0.9247, 0.8959, 0.8511, 0.6814, 0.9987, 0.8271, 0.7521, 0.6764, −0.8862, 0.5856, −0.6921, 0.0793, and 0, respectively. In combination, the two axes explain 86.31% of the composition change in species. Altitude, latitude, longitude, TN, TP, SOC, SMC, pH, Max T, AT, Min T, SD, DTR, precipitation, ARH, and WS determined the distribution of the community of species, and the respective coefficients (R2 values) were 0.3509, 0.3019, 0.5246, 0.2724, 0.1754, 0.1801, 0.1144, 0.176, 0.1674, 0.1442, 0.1399, 0.0179, 0.0054, 0.0077, 0.0705, and 0.2546. The order of influence on the root fungal community was longitude > altitude > latitude > TN > WS > SOC > PH > TP > Max T > AT > Min T > SMC > ARH > SD > precipitation > DTR. Thus, the geographical factors longitude, altitude, and latitude exerted the greatest influence on the root fungal community of K. humilis.

Figure 7 CCA analysis of main genera of fungi community (top 10) and environmental factors in root and rhizosphere soil of K. humilis.

For the rhizosphere soil samples, the first CCA axis (CCA1) explained 51.81% of the variation (Fig. 7B), and the correlation coefficients with altitude, latitude, longitude, TN, TP, SOC, SMC, pH, Max T, AT, Min T, SD, DTR, precipitation, ARH, and WS were 0.9928, −0.9786, −0.1239, −0.3194, 0.8406, −0.3907, 0.9908, −0.9478, −0.5966, 0.4078, 0.5339, −0.989, −0.5584, 0.9894, −0.8566, and 0, respectively. The second axis (CCA2) explained 22.73% of the variation, and the correlation coefficients with altitude, latitude, longitude, TN, TP, SOC, SMC, pH, Max T, AT, Min T, SD, DTR, precipitation, ARH, and WS were −0.1198, −0.2059, 0.9923, −0.9476, −0.5416, −0.9205, −0.1355, −0.3187, −0.8025, 0.9131, 0.8455, −0.1481, −0.8296, −0.1452, 0.516, and 0, respectively. Combined, the two axes explained 74.54% of the composition change in species. Altitude, latitude, longitude, TN, TP, SOC, SMC, pH, Max T, AT, Min T, SD, DTR, precipitation, ARH, and WS determined the distribution of community species, with respective coefficients (R2 values) of 0.0901, 0.2517, 0.181, 0.1208, 0.169, 0.0976, 0.1291, 0.4788, 0.0568, 0.0073, 0.0954, 0.0082, 0.1708, 0.088, 0.0187, and 0.0133. The order of influence on the root fungal community was pH > latitude > longitude > DTR > TP > SMC > TN > SOC > Min T > altitude > precipitation > Max T > ARH > WS > SD > AT. Thus, pH was the most important environmental factor affecting the fungal community in the rhizosphere soil of K. humilis.

Spearman rank correlation coefficients between the environmental factors and the microflora were used to produce a heatmap (Fig. S4), revealing that 53 and 55 genera significantly correlated with the environmental factors in root and rhizosphere soil, respectively. Various fungi were affected by environmental factors. In root samples, SMC, TP, TN, and SOC were significantly positively correlated with the fungal genera Microsporomyces, Plenodomus, Marasmiellus, and Coprinus. Latitude and pH were significantly positively correlated with the genera of Vishniacozyma, Naganishia, Efibulobasidium, Monocillium, Fusariella, Podospora, Gibberella, and Filobasidium. Additionally, 10 and 16 genera, respectively, significantly correlated with longitude and altitude. In the rhizosphere soil samples, pH, latitude, and longitude were significantly positively correlated with the genera Floccularia, Microdochium, Hygrocybe, Lapidomyces, Pgrenochaeta, Pulvinula, Neostagonospora, Cistella, Comoclsthris, Neosetophoma, Neoascochyta, and Stagonospora. Furthermore, six and three genera, respectively, were significantly positively correlated with the soil properties and altitude. In addition, 22 genera exhibited a significant correlation with environmental parameters.

Discussion

The K. humilis rhizosphere is rich in fungi

This study used Illumina Miseq to assess fungal distribution in the rhizosphere of K. humilis in the alpine meadow of the QTP; we identified 19,423 and 25,101 fungal OTUs from the roots and rhizosphere soil, respectively. There were 435 core OTUs identified in the roots and 415 in the rhizosphere soil, which is higher than that of Cyperus esculentus (40 core OTUs) (Wang et al., 2022a) and Oxytropis glacialis (37 core OTUs) (Cao et al., 2022). This indicates the rich fungal diversity of the K. humilis rhizosphere. In this study, we found 68 core genera in the roots and 59 core genera in the rhizosphere soil, of which 25 were common to both. For example, there was a relative abundance >1% for Mortierella, Microscypha, Floccularia, Cistella, Gibberella, and Pilidium. Thus, the K. humilis root system is highly developed, which has resulted in a stable core fungal community developing during its long-term evolution. The 25 core fungal communities found in this research are of great significance for studies of the adaptative evolution of Kobresia species, such as K. humilis in the alpine meadows of the QTP. Mortierellaceae, Agaricaceae, and Nectriaceae were found to play a key role in the construction of the metacommunity-scale network, which may help to reveal the adaptive evolutionary mechanism of the fungal community of K. humilis and the role it plays in extreme environments.

Differences in the fungal community of K. humilis roots and rhizosphere soil

We found that fungal community diversity and composition differed between the roots and the rhizosphere soil of K. humilis. The diversity, abundance, and total number of OTUs in the rhizosphere soil of K. humilis was higher by 11.85%, 9.85%, and 22.62%, respectively, than those of endophytic fungi in the root. This may be because the rhizosphere soil serves as a bridge between the internal and external habitats of plants, acting as a microenvironment through which plants interact with the outside environment. Roots of different individuals and species, roots and insects, and roots and microorganisms all interact with one another in the rhizosphere soil. These intricate interactions influence the diversity of soil microorganisms in the rhizosphere (Yu et al., 2012). The interior environment of plant roots, in contrast, is more stable and uniform than the rhizosphere, leading to a more uniform population of endophytic fungi. Overall, the adaptability of plants and microorganisms is responsible for both the diversity of fungal communities in rhizosphere soil and the relative simplification of endophytic fungal communities (Bais, Weir & Perry, 2006; Liu et al., 2017a, 2017b; Wang et al., 2022b). In similar studies by Arslan et al. (2020), Crgger, Veach & Yang (2018), and Zarraonaindia, Owens & Weisenhorn (2015), fungal diversity was found to decrease in the root but increase in the rhizosphere soil. However, this is inconsistent with the findings of Egan et al. (2016) and Bahram et al. (2012), who claimed that root fungal diversity was higher than that of rhizosphere soil. Indeed, we found that the fungal diversity in roots was higher than the inter-root fungal diversity at some sampling sites, such as ZK. This may be related to the high diversity and abundance of plant species found by Mao et al. (2022) at ZK.

In addition, in the eight areas of the QTP, there were differences in the composition and diversity of fungal communities. First, Ascomycota was the most common phylum in all areas. Basidiomycota was the second most abundant phylum in HY, ZK, and XH (Fig. 3A), and Mortierellomycota was second in TJ, QSH, ZQ, MQ, and GD (Fig. 3D). Second, diversity indices of roots and rhizosphere soil showed relatively high diversity at the GD (3,960 m a.s.1.) and QSH (4,410 m a.s.1.) sites, respectively. The reasons for the above differences may be related to the different types of litter present because of the different vegetation communities (Jiang, Xu & Xu, 2006). Previous studies have shown that members of Basidiomycota mostly rely on exogenous substances such as plant litter or soil organic matter as the main carbon source for growth and reproduction. They can decompose refractory lignin and occupy an important position in forests with high lignin content litter (Bossuyt, Denef & Six, 2011). As a unique group, Mortierellomycota may play a role in promoting material cycling in enclosed grasslands with low nutrient content (Veach et al., 2018). This may also be related to the high diversity and abundance of plant species found by Mao et al. (2022) in the areas of ZK, HY, and XH.

Fungal ecological types in the roots and rhizosphere soil of K. humilis

According to our results, the saprotroph-symbiotroph mode represents the main ecological function of fungi in the roots and rhizosphere soil. There are few pathotrophs in the roots, whilst there were more pathotrophs and pathotroph-saprotrophs in the rhizosphere soil. This could be due to the roots’ long-term close contact with the soil, which allows more saprophytic fungi to enter the roots and become endophytic fungi. It could also be related to the formation of root fungal communities as a result of host plants filtering and selecting fungi in the soil (Wang et al., 2020). Generally, symbiotrophic fungi are extremely beneficial to the health, nutrition, and quality of plants (Igiehon & Babalola, 2017). Pathotrophic fungi generally obtain nutrients by attacking host cells; thus, they are considered to cause disease or exert negative effects on plant performance (Anthony, Frey & Stinson, 2017). Furthermore, we discovered that the proportion of pathotroph fungi in K. humilis root samples in the QSH, ZQ, MQ, and GD areas was lower than in the HY, ZK, XH, and TJ areas. The reason may be that QSH, ZQ, MQ, and GD are located at high altitudes, 3,900–4,410 m, and are in remote regions with less human interference. However, HY, ZK, XH, and TJ are situated at altitudes of 3,200–3,600 m in areas with a high human influence, and there are numerous human activities that may cause physical damage to plants, infect plants with fungal diseases, and increase the presence of plant root pathotrophs. In addition, the ecological function of most fungi in the roots and rhizosphere soil of K. humilis is still unknown, indicating that the functions of roots and rhizosphere fungi are complex and have great development potential. In the future, it would be valuable to carry out isolation and cultivation of fungal communities in the K. humilis ecosystem, and further study their functional characteristics.

Longitude and pH were the most important factors driving the fungal community in K. humilis

For fungi, the growth environment is crucial for the selective colonization. In this study, the environment of the eight regions of the Qinghai-Tibet Plateau varies greatly, with an altitude range of 3,200–4,410 m, a longitude range of 97°9′24″–101°28′1″, and a latitude range of 33°23′34″–37°13′40″. However, there are 25 genera of core fungi in the root and rhizosphere soils of K. humilis in all eight regions, of which the top ten fungi in terms of relative abundance are Mortierella, Gibberella, Floccularia, Didymella, Cistella, Mycosphaerella, Microscypha, Preussia, Gymnoascus and Stagonospora. The distribution of these core fungi may not be affected by environmental factors, and the macro environment of the Qinghai-Tibet Plateau and the microenvironment of K. humilis may determine their distribution. In addition, the eight regions also support unique fungi, for example, Periconiella, Rhizophlyctis, Talaromyces, Strelitziana, and Heteroconium were only recorded in HY; Flavomyces, Coniothyrium, and Marchandiobasidium only in ZK; Geomyces, Phaeoisaria, Paulisebacina, and Leucosporidium only in XH; Tubaria, Neodevriesia, Coriolopsis, Kurtzmanomyces only in TJ; Helvellosebacina, Monosporascus, Podospora, Clavulina, Heyderia, Niesslia, and Gaeumannomyces only in QSH; Herpotrichia and Mollisina only in ZQ; Angustimassarina, Hemileucoglossum, Trichophaeopsis, Tolypocladium, Colletotrichum, and Oidiodendron only in MQ; Metschnikowia only in GD. This may be due to environmental factors such as longitude, latitude, altitude, and soil pH. According to other studies, plant species occur along strong altitude or latitude gradients and with corresponding changes in community structure (Daco, Colling & Matthies, 2021). In this study, we analyzed the relationship between the environmental factors and the fungal community in the root and rhizosphere soil of K. humilis. We found that longitude, altitude, and latitude were significantly associated with the fungal community structure in the roots, which was similar to the results by Sa, An & Sa (2012), Yao et al. (2021), and Arslan et al. (2020). Furthermore, many studies have shown that fungal community structure is driven by soil physical and chemical characteristics (Schmidt et al., 2011; Yuan et al., 2014; Cao et al., 2022); in particular, soil pH is a key factor controlling fungal community structure (Sui et al., 2022; Lin et al., 2021; Liu et al., 2022). Interestingly, we found that soil pH was the most critical factor for the fungal community structure of the rhizosphere soil. This finding is consistent with previous observations that the fungal community significantly changes the soil pH (Li et al., 2020). In summary, these results reveal that the rhizosphere community structure of K. humilis could be determined by environmental changes.

Conclusion

The fungal community structure of K. humilis roots and the rhizosphere soil differs significantly across eight areas of the QTP as a result of the long-term evolutionary process. The diversity, abundance, and total number of OTUs in K. humilis rhizosphere soil were greater than those of the endophytic root fungi. The main ecological types of fungi in the roots and rhizosphere soil were saprotroph-symbiotrophs. The roots had few pathotrophs, whereas rhizosphere soil had more pathotrophs and pathotroph-saprotrophs. Longitude and pH were the most important factors controlling the fungal community in K. humilis roots and rhizosphere soils, respectively. This study is the first to show the fungal distribution of K. humilis on the QTP. Our data provide a reference for the sustainable utilization of biological resources on the QTP.

Supplemental Information

Supplemental Information 1 Supplementary Tables.

Supplemental Information 2 Supplementary figure.

Additional Information and Declarations

Competing Interests

Author Contributions

Data Availability

The authors declare there are no competing interests.

Jing Guo conceived and designed the experiments, performed the experiments, analyzed the data, prepared figures and/or tables, and approved the final draft.

Zhanling Xie performed the experiments, prepared figures and/or tables, and approved the final draft.

Qing Meng analyzed the data, prepared figures and/or tables, authored or reviewed drafts of the article, and approved the final draft.

Hongyan Xu analyzed the data, prepared figures and/or tables, and approved the final draft.

Qingqing Peng analyzed the data, authored or reviewed drafts of the article, and approved the final draft.

Bao Wang performed the experiments, analyzed the data, authored or reviewed drafts of the article, and approved the final draft.

Deyu Dong performed the experiments, analyzed the data, authored or reviewed drafts of the article, and approved the final draft.

Jiabao Yang performed the experiments, analyzed the data, authored or reviewed drafts of the article, and approved the final draft.

Shunbin Jia performed the experiments, authored or reviewed drafts of the article, and approved the final draft.

The following information was supplied regarding data availability:

The data is available at NCBI: SAMN34510363 and SAMN34510364.

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
