# Peer review of "Distribution of rhizosphere fungi of Kobresia humilis on the Qinghai-Tibet Plateau"

_PeerJ, doi:10.7717/peerj.16620_

## Round 0.1 · original submission · Major Revisions

The authors should be improved the language. The authors should describe all the methods and discussion in detail.

Reviewer 1 ·

Basic reporting

This article showed the fungal distribution of K. humilis on the QTP during the long-term evolutionary process. These findings are critical for determining the effects of environmental variables on K. humilis fungal communities and can provide valuable guidance for ecological restoration and sustainable utilization of biological resources of the QTP.

Experimental design

L120: In Materials and methods, you showed that “Each sample point was set with twenty repeats, and the distance between the twenty repeats was above 20 m”, but why in Figure 3 and Figure 7, there are less than 20 replicates per sampling point, and three replicates were shown in Table S5, please describe the specific method of analysis.

Validity of the findings

no comment

Additional comments

This has a couple of issues that need to be modified

1. Although in the Manuscript marked the root and rhizosphere, but it is necessary to sign in the title of figures like figure 6. For example, Figure 2 Fungal community diversity index of root (A, B and C) and rhizosphere soil (D, E and F). Figure 3, 4, 5, and 7 should also modify.
2. Figure 1B only showed one region of soil samples, if you think it’s nessessary to show the picture of soil samples of regions, you need show all picture of regions.
3. L126: Determination of rhizosphere soil physicochemical index showed the method of TN、TP、TOC and pH,but do not show where you got other Environmental factor information (Max T: daily maxium temperature; A T: average temperature; Min T: daily minimum temperature; SD: sunshine duration; DTR: daily temperature range; ARH: average relative humidity; WS: wind speed.).
4. L322: “latitude” and “longitude” should be “Latitude” and ‘Longitude”.
5. In the discussion, each subheading should be a concluding sentence and should not be similar to the outcome subheading. For example, ”Relationship between environmental factors and fungal community structure” should be “Longitude and pH were
the most important factors driving the fungal community in K. humilis”.
6. L361: There were 435 core OTUs identified in the root and 415 core OTUs identified in the rhizosphere soil, which was higher than that of Cyperus esculentus (Wang et al, 2022) and Oxytropis glacialis (Cao et al, 2022). You should show the number of Cyperus esculentus and Oxytropis glacialis.
7. The discussion does not explain why the fungal diversity in roots is higher than the inter-root fungal diversity in some sampling sites, the explanation is too general.
8. L418: “HY ZK XH” should be “HY, ZK, and XH”.
9. It needs to be checked from scratch for grammar and logic, and the overall level of the manuscript leaves something to be desired.

Reviewer 2 ·

Basic reporting

Kobresia humilis is a main constructive species and plays crucial roles in maintaining the ecological balance in the alpine meadow of the QTP. As we know, rhizosphere fungi contributed essential living habitats to the host plants. The unique environment of QTP provide us a special ecological environment to study adaptive evolution. Hence, Guo et al. conducted a comprehensive analysis for the distribution of rhizosphere fungi of K. humilis on the QTP to study the adaptive mechanism underlying the long-term evolutionary process. However, it can’t be accepted before the following issues addressed.

Experimental design

1. The authors emphasized on the comparison of rhizosphere fungal community structure of K. humilis distributed in the QTP, but it seems that the authors only collected samples from the eastern part of Qinghai Province. Please explain what principles were followed in the design of the sampling sites, whether they were limited by objective conditions or by plant distribution areas.
2. The number of biological replicates per sampling site used to analyze fungal community structure needs to be specified in the M&M or results sections. The M&M section indicates 20 replicates, but Figure 3 does not seem to be, which is very confusing.
3. The bioinformatics and data statistical analyses section need to cite inferences when some methods or software used.

Validity of the findings

4. I am less convinced that longitude can be a main factor driving structural differences in rhizosphere fungi of plants, unless the authors can present more direct evidence to support this conclusion.
5. Line 176-181: The number of ITS sequences varies greatly between rhizosphere fungi and root fungi. Is it the depth of sequencing or the amount of fungi that causes this phenomenon? The first two sentences do not seem to draw the conclusion of the third sentence, please reconsider it carefully.
6. What are the differences in K. humilis phenotypes among different sampling sites? Throughout the MS, it seems that the authors have only compared the differences among different sampling sites, but have not shown how these differences relate to long-term evolution. If the author wants to keep the current title, I suggest that the authors try to explain the underlying mechanism of community structure change from an evolutionary perspective.

---

## Round 0.2 · Major Revisions

Language quality is very poor. The authors should improved the language of the manuscript and describe in detail the material and methods and discussion section.

**Language Note:** The Academic Editor has identified that the English language must be improved. PeerJ can provide language editing services - please contact us at [email protected] for pricing (be sure to provide your manuscript number and title). Alternatively, you should make your own arrangements to improve the language quality and provide details in your response letter. – PeerJ Staff

---

## Round 0.3 · accepted · Accept

The authors have revised the manuscript as per the comments. So, I suggest accepting the manuscript for possible publication.